# CALANet: Cheap All-Layer Aggregation for Human Activity Recognition

**Jaegyun Park[1], Dae-Won Kim[1,*], Jaesung Lee[2,*]**
[1]School of Computer Science and Engineering, Chung-Ang University, Republic of Korea
[2]Department of Artificial Intelligence, Chung-Ang University, Republic of Korea
jgp0566.cau@gmail.com, {dwkim, curseor}@cau.ac.kr

## Abstract

With the steady growth of sensing technology and wearable devices, sensor-based human activity recognition has become essential in widespread applications, such as healthcare monitoring and fitness tracking, where accurate and real-time systems are required. To achieve real-time response, recent studies have focused on lightweight neural network models. Specifically, they designed the network architectures by restricting the number of layers shallowly or connections of each layer. However, these approaches suffer from limited accuracy because the classifier only uses the features at the last layer. In this study, we propose a cheap all-layer aggregation network, CALANet, for accuracy improvement while maintaining the efficiency of existing real-time HAR models. Specifically, CALANet allows the classifier to aggregate the features for all layers, resulting in a performance gain. In addition, this work proves that the theoretical computation cost of CALANet is equivalent to that of conventional networks. Evaluated on seven publicly available datasets, CALANet outperformed existing methods, achieving state-of-the-art performance. The source codes of the CALANet are publicly available at `https://github.com/jgpark92/CALANet`.

## 1 Introduction

Human activity recognition (HAR) is a fundamental technique in healthcare [28, 53], fitness tracking [5, 23], and surveillance [21, 41]. Wearable sensor-based HAR has drawn attention in pervasive computing applications due to the popularity of smart wearable devices in recent years [7, 53]. Specifically, it aims to identify motion details of users or activity tracks from sensor signal patterns [10]. To this end, neural networks (NNs) have been widely used to achieve a superior learning performance without handcrafted feature engineering [7, 52]. Besides, with advances in microelectronics and inertial sensor-based wearable devices, recent researchers have focused on achieving real-time systems [4, 43]. Especially, a recent trend across most studies has become increasingly to train NNs on a resource-rich computing device and then deploy them to resource-limited wearable devices, where inference is executed [44, 60, 61].

Recent real-time HAR studies have focused on one-dimensional (1D) convolutional neural networks (CNNs) compatible with various hardware accelerators and deployment frameworks [14, 20, 23, 36, 37, 48, 51, 60]. CNNs include convolution and pooling operations, which allow them to extract more abstracted high-level features as input signals pass from early to later layers. Specifically, the pooling operation abstracts signals by reducing the feature size (temporal resolution), which can be regarded as a sampling of signal. As a result, the final classifier predicts an activity class based on the abstracted or sampled features at the last layer. The sampled features are more semantic and global

---

*Corresponding authors.

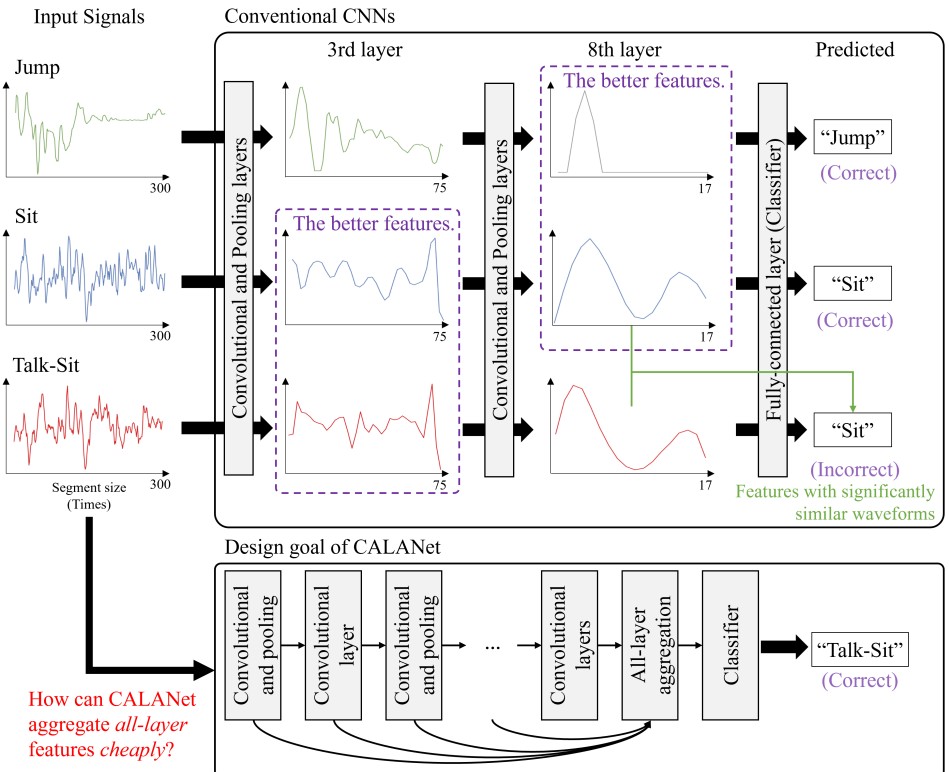

Figure 1: Analysis of representations in our experiments on KU-HAR dataset [47]. In a conventional CNN, the classifier predicts activities only using the feature representations at the last layer. Features at the early layer include the detailed information of original signals that may confound the classifier. In comparison, features at the later layer are more semantic, but the features (with more compact and short waveforms) make it challenging to classify activities that share similar semantics. Our goal is to design a CALANet that allows the classifier to use features for all layers while maintaining the inference time of conventional CNNs.

than those of prior layers, but it has not been proven that the last layer is the optimal representation [58]. Although the high-level features sampled by the pooling operation can avoid over-fitting of the classifier [59], for the HAR dataset, the loss of some detailed information makes it challenging to classify activities that share similar semantics, such as "Sit" and "Talk-Sit."

Figure 1 illustrates intermediate features in the forward pass of conventional CNNs. Conventional CNNs classify "Jump" and "Sit" well but tend to misclassify "Talk-Sit" as another activity, as shown in Figure 1. These experimental observations have also been reported in existing studies [24, 47]. Specifically, the features at the eighth layer (with more compact and short waveforms) make it easy for the classifier to discriminate "Jump" and "Sit," compared with the ones at the third layer. On the contrary, the features at the third layer (with more detailed information) can be more suitable than the ones at the eighth layer when classifying "Sit" and "Talk-Sit" that have similar vibrations in signal waveforms. Although which layer has the best features depends on the activity, conventional CNNs classify multiple activities only using the features at the last layer.

The objective of real-time HAR is to maximize accuracy under real-time constraints. To achieve real-time response, prior HAR studies designed the network architectures of CNNs by restricting the number of layers shallowly [20, 23, 51, 60] or reducing connections of each layer [14, 36, 37, 48]. However, these approaches suffer from limited accuracy because their classifier only uses the features at the last layer. A straightforward approach to address this issue is to allow the classifier to use the features for all layers [19, 27], but this leads to a substantial increase in computational cost, particularly as the number of layers deepens. Therefore, our goal is to design a novel network architecture that allows the network to aggregate the features for all layers *while maintaining the computational cost of the conventional CNNs regardless of network depth*, as shown in Figure 1.

In this paper, we propose a novel network, CALANet, with a cheap *all-layer* aggregation (CALA) structure. To achieve our goal, CALANet includes (1) learnable channel-wise transformation matrices and (2) scalable layer aggregation pool. First, we introduce new learnable channel-wise transformation matrices (LCTMs) to minimize an increase in computational cost due to all-layer aggregation. Given intermediate features at a specific layer (with temporal resolution $T$ and the number of channels $M$), $M$ LCTMs generate a vector with $N \ll T$ elements based on linear transformation and combination without increasing the theoretical computation cost. Second, we improve the effectiveness of all-layer aggregation by introducing a scalable layer aggregation pool (SLAP) that allows CALANet to stack layers without significantly increasing computational costs. As a result, the main contributions of this paper are as follows:

- We proposed CALANet with a CALA structure that allows the network to aggregate the features for all layers while maintaining the efficiency of CNNs regardless of network depth based on (1) LCTMs and (2) SLAP.
- We theoretically proved that the computational cost of CALANet is equivalent to that of conventional CNNs, including even shallow networks.
- We empirically demonstrated the effectiveness of CALANet in achieving superior performance compared to 11 state-of-the-art methods on seven public benchmark datasets.

## 2 Related Work

Many comprehensive surveys in HAR literature have highlighted the importance of NN-based models and real-time applications [7, 10, 34, 42–44, 52, 61]. For instance, early real-time HAR studies adopted two-dimensional (2D) convolutional NNs (CNNs) with shallow architectures [9, 25, 33, 45]. Specifically, they transformed the sensor signal patterns to 2D spectral images as an input of 2D CNNs. However, these approaches require complex preprocessing, such as discrete or short-time Fourier transform, which increases the overhead during continuous processing for real-time HAR. Meanwhile, Ignatov [23] proposed a one-dimensional (1D) CNN architecture using basic statistical features to encode global temporal information. Although this approach demonstrated the potential of 1D CNN, it still requires several data preprocessing like calculating the histogram of input signals.

Recent real-time HAR studies focus on 1D CNNs without any complex data preprocessing. For example, Zebin et al. [60] proposed a CNN architecture comprising four convolution layers. Furthermore, they showed the efficiency of parameter quantization as post-processing for further optimization. In another study, Wan et al. [51] adopted a CNN architecture including three convolution layers. They demonstrated the superiority of CNNs compared with recurrent NN variants on HAR datasets with basic activities. To enhance shallow CNNs, Huang et al. [20] introduced a cross-channel communication that exchanges information among channels within the same layer. These models achieved real-time HAR by shallowly restricting the number of layers, resulting in limited accuracy.

To alleviate the issue, some studies have considered efficient variants of convolution at each layer instead of reducing the number of layers. For example, Gao et al. [14] proposed a selective kernel module that divides the convolution into split, fuse, and select steps to adjust receptive field size adaptively. Similarly, Tang et al. [48] designed a hierarchical-split block to enhance multiscale temporal features by composing channel groups hierarchically. In another study, Teng et al. [49] proposed RepHAR, which re-parameterizes a pretrained multibranch CNN to a plain CNN before deploying it into resource-limited devices. However, these approaches still are insufficient to classify activities that have similar vibrations in signal waveforms because the classifier only uses the features at the last layer. Meanwhile, Park et al. [37] introduced a grouped temporal shift network that can flexibly re-design a network architecture to support various hardware specifications. Although this network can derive layer-specific structures suitable for a given computational budget, its performance is limited according to an initial network architecture. Therefore, its performance can be improved by using our CALANet as the initial network, as will be described in Section 4.2.

Besides, the classifier requires both local and global temporal representations to achieve high HAR accuracy [64]. Specifically, the locality of CNNs improves accuracy due to their translational invariance concerning the precise location of activity within a segment of time-series data [17]. On the other hand, recurrent layers or attention mechanisms have an advantage for global feature extraction because they can model long-term dependencies. In this regard, many studies have attempted to integrate recurrent layers [8, 24, 35, 50, 56] or attention mechanisms [40, 63] into

CNNs, which has increased both accuracy and inference times. The increase in inference time is primarily because of the lack of device-level optimizations compared with CNNs [15, 29, 60]. These accuracy-oriented networks will be compared with our CALANet in Section 4.2.

## 3 Cheap All-Layer Aggregation Network

In this section, our goal is to design a CNN architecture that can aggregate features for all layers into the final classifier without increasing the computational cost of CNNs. Furthermore, we prove that the theoretical computation cost of the proposed CALANet is equivalent to that of conventional CNNs, including even shallow networks.

### 3.1 Computational cost of convolutional neural networks

Our goal is to improve the HAR accuracy while maintaining the efficiency of CNNs. Therefore, before deriving a novel network structure and proving its theoretical efficiency, we formalize the computational cost of the conventional CNNs in a generalized form. Note that we only consider the computational cost in the feed-forward step, not the training step, which is unrelated to inference time. Let $\mathcal{K}^{(l)}$ be the $l$-th layer of a network, where $\mathcal{K}^{(l)}$ and $\mathcal{K}^{(l+1)}$ are calculated sequentially and independently. Therefore, the theoretical computation cost of the CNN can be defined as a summation for each computation of layers, as described in Definition 1.

**Definition 1.** *Let $\alpha(\cdot)$ be the computational cost of calculating the output of each layer. Given a network architecture $\mathcal{A}$ with $L$ layers, the input $X^{(l)}$ is fed into the $l$-th layer with trainable parameters $\theta^{(l)}$ to calculate the output $X^{(l+1)}$. Due to this layer composition, its computational cost is formalized as*

$$C_n(\mathcal{A}) = \sum_{l=1}^{L} \alpha(\mathcal{K}^{(l)}(X^{(l)}; \theta^{(l)})). \tag{1}$$

To formalize the computation cost of CNNs, we borrow the concept of time complexity as the upper bound of the computational cost. Similar to Proposition 3 of [37], Eq. (1) is simplified by Proposition 1.

**Proposition 1.** *The time complexity of CNNs is formalized as:*

$$\mathbb{M} \le \mathbb{N}(L-1) \implies \mathcal{O}(\mathbb{T}\mathbb{D}_k\mathbb{N}^2 L). \tag{2}$$

*where $\mathbb{M}$ and $\mathbb{T}$ are the number of channels and temporal resolution for input data, and $\mathbb{N}$ and $\mathbb{D}_k$ are average numbers for output channels and kernel sizes across a network, respectively.*

The proof is given in Appendix A. Because we do not restrict the number of layers shallowly, we will assume that the condition of Eq. (2) is always true in this paper.

### 3.2 Learnable channel-wise transformation matrix

In this section, we introduce a learnable channel-wise transformation matrix (LCTM) that allows our CALANet to aggregate features for all layers without increasing the theoretical computation cost. Figure 2 shows the network architecture of CALANet. Given input signals, the convolution and pooling layers extract the high-level sampled features by calculating temporal correlations and reducing the feature resolution. After that, the features for all layers are connected with the classifier via the cheap all-layer aggregation module based on the LCTMs.

Let $\mathbf{x_m}$ be a feature vector of $m$-th channel with temporal resolution $T = |\mathbf{x_m}|$ at any layer. Because $T$ varies with the layer, we define a mapping function $f : \mathbf{x_m} \to \mathbf{y} \in \mathbb{R}^N$, where a constant value $N \ll T$. After that, we calculate $f$ via a transformation matrix $\mathbf{A} \in \mathbb{R}^{N,T}$. As the mapping function is calculated for each channel of features, $m$-th feature vector is transformed to $\mathbf{y_m} \in \mathbb{R}^N$ as follows:

$$\mathbf{y_m} = \mathbf{A_m}\mathbf{x_m} \tag{3}$$

where this transformation can be interpreted as a compression of global temporal information. After that, a linear combination is conducted to calculate relations between channels as follows:

$$\hat{\mathbf{y}} = \sum_m a_m \mathbf{y_m} \tag{4}$$

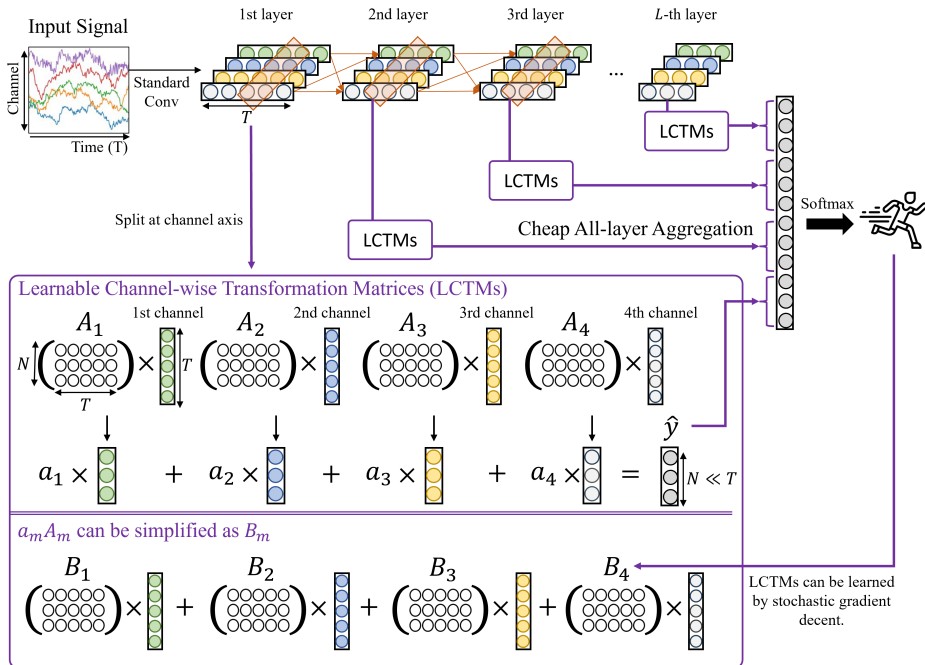

Figure 2: Network architecture of CALANet. Convolution and pooling layers extract the sampled features by reducing the temporal resolution. CALANet aggregates the features for all layers based on the linear transformation and combination.

where $\hat{\mathbf{y}} \in \mathbb{R}^N$ is a vector and $a_m$ is $m$-th coefficient of the linear combination.

We replace $a_m \mathbf{A_m}$ with $\mathbf{B_m} \in \mathbb{R}^{N,T}$. As a result, Eq. (4) is simplified as follows:

$$\hat{\mathbf{y}} = \sum_m \mathbf{B_m} \mathbf{x_m} \tag{5}$$

where elements of $\mathbf{B_m}$ can directly be optimized by stochastic gradient descent because these matrix multiplications can be implemented by dense layers. Therefore, we define $B_m$ as LCTM. Herein, the time complexity of calculating $M$ LCTM operations is $\mathcal{O}(TMN)$. Finally, $\hat{\mathbf{y}}$ for all layers are concatenated and then fed into the classifier.

To investigate whether our CALANet can maintain the efficiency of conventional CNNs, we analyze the time complexity of CALANet. Based on Proposition 1, the time complexity of CALANet is formalized in Lemma 1.

**Lemma 1.** *The time complexity of CALANet is equivalent to:*

$$\mathcal{O}(\mathbb{T}\mathbb{D}_k\mathbb{N}^2 L). \tag{6}$$

The proof is given in Appendix B. According to Proposition 1 and Lemma 2, our CALANet can aggregate features for all layers while maintaining the efficiency of conventional CNNs.

### 3.3 Scalable layer aggregation pool

The effectiveness of all-layer aggregation depends on a layer aggregation pool, i.e., the number of layers, as shown in Table 3. To improve the accuracy of CALANet further, we also introduce a scalable layer aggregation pool (SLAP) that allows CALANet to stack layers without significantly increasing computational cost. To this end, we, in this section, aim to omit $L$ in Eq. (2). Inspired by ShuffleNet [62], we first use the grouped convolution and channel shuffle to reduce the time complexity of the standard convolutions. Precisely, the M input channels are evenly divided into $G$ channel groups. After that, the standard convolution generates $\lfloor N/G \rfloor$ output channels for each channel group. Subsequently, the channel shuffle operation is executed.

The entire channels are fully related by the channel shuffle operations if and only if (the number of layers) $\times$ (the number of channels within each channel group) $\geq$ (the number of channel groups) [62]. Therefore, $G$ is set into a value satisfying $L \times N \geq G^2$. As $N \geq G$ and $G$ are inversely proportional to the computational cost, we set $G$ into $L$ without any loss of information for channel correlations. Consequently, the time complexity of the stack of the standard convolutions is reduced in Lemma 2.

**Lemma 2.** *The time complexity of calculating the standard convolutions is reduced to:*

$$\mathcal{O}(\mathbb{T}\mathbb{D}_k\mathbb{N}^2). \tag{7}$$

The proof is given in Appendix C. To reduce the time complexity of all-layer aggregation, we focus on the norm of vectors extracted from LCTMs, i.e., $|\hat{\mathbf{y}}| \approx \mathbb{N}$ from Eq. (5). The LCTMs for all layers generate a vector with $L \times \mathbb{N}$ elements fed into the Softmax layer to classify the activities. The large number of units in the Softmax layer may incur overfitting [57]. Therefore, we fix the number of features fed into the Softmax layer to $\mathbb{N}$ by dividing the number of rows of LCTM in Eq. (5) by $L$, resulting in $\mathbf{B_m} \in \mathbb{R}^{N/L,M}$. Consequently, the time complexity of all-layer aggregation is reduced in Lemma 3.

**Lemma 3.** *The time complexity of calculating all-layer aggregation is formalized as:*

$$\mathcal{O}(\mathbb{T}\mathbb{N}^2). \tag{8}$$

The proof is given in Appendix D. Finally, we introduce CALANet with the SLAP by omitting the factor $L$ from its time complexity. Consistent with Lemma 2 and Lemma 3, the time complexity of CALANet is reduced in Theorem 1.

**Theorem 1.** *The time complexity of CALANet is reduced to:*

$$\mathcal{O}(\mathbb{T}\mathbb{D}_k\mathbb{N}^2). \tag{9}$$

The proof is given in Appendix E. From Theorem 1, we crosscheck the efficiency of our CALANet by making comparisons with the time complexity of shallow CNNs in Corollary 1 and Corollary 2.

**Corollary 1.** *The time complexity of CALANet is equivalent to the shallow CNNs with $L \geq 2$.*

*Proof.* The time complexity of shallow CNNs with $L = 2$ is $\mathcal{O}(\mathbb{T}\mathbb{D}_k\mathbb{M}\mathbb{N}) + \mathcal{O}(\mathbb{T}\mathbb{D}_k\mathbb{N}^2) = \mathcal{O}(\mathbb{T}\mathbb{D}_k\mathbb{N}^2)$. It is equivalent to Eq. (9). $\qquad \square$

**Corollary 2.** *The time complexity of CALANet is equivalent to the shallow CNNs with $L = 1$ if $\mathbb{M} \approx \mathbb{N}$.*

*Proof.* The time complexity of shallow CNNs with $L = 1$ is $\mathcal{O}(\mathbb{T}\mathbb{D}_k\mathbb{M}\mathbb{N})$. If $\mathbb{M} \approx \mathbb{N}$, then it is equivalent to Eq. (9). $\qquad \square$

In conclusion, our CALANet has a computation cost comparable to shallow CNNs. Especially from Corollary 2, the time complexity of CALANet becomes equivalent to the shallow CNNs even with $L = 1$ as the number of sensors increases.

## 4  Experiments

In this section, we evaluate the superiority of CALANet. In Section 4.1, we describe the experimental setup. Section 4.2 presents the compared results of CALANet and other networks on seven HAR datasets. Section 4.3 provides an in-depth analysis via an ablation study. Lastly, Section 4.4 measures the actual inference time of CALANet.

### 4.1  Experimental Settings

**Dataset.** We used seven public benchmark datasets, including various sampling frequencies, the number of activities, and sensors. They include **UCI-HAR** [1], **UniMiB-SHAR** [30], **DSADS** [3], **OPPORTUNITY** [6], **KU-HAR** [47], **PAMAP2** [46], and **REALDISP** [2]. The details for each dataset are described in Appendix F.

Table 1: Comparison results on seven datasets. ▼/△ indicates that the corresponding model is significantly worse/better than CALANet according to a paired $t$-test at a 95% significance level.

| Model | UCI-HAR | | UniMiB-SHAR | | DSADS | | OPPORTUNITY | |
|---|---|---|---|---|---|---|---|---|
| | F1 | FLOPs | F1 | FLOPs | F1 | FLOPs | F1 | FLOPs |
| CALANet (**Ours**) | 96.1 | **7.6M** | 78.3 | **8.8M** | 90.0 | **8.5M** | 81.6 | **19.3M** |
| CALA-GTSNet (**Ours**) | 94.7▼ | 3.3M | 74.1▼ | 4.8M | 87.2▼ | 5.4M | 78.4▼ | 15.0M |
| Shallow ConvNet [23] | 92.5▼ | 17.9M | 72.2▼ | 18.2M | 85.6▼ | 48.5M | 79.5▼ | 74.3M |
| RepHAR [49] | 95.1▼ | 31.8M | 71.6▼ | 37.3M | 85.5▼ | 32.9M | 80.0▼ | 26.0M |
| Res-GTSNet [37] | 94.5▼ | 6.4M | 77.2▼ | 7.51M | 84.4▼ | 7.3M | 76.0▼ | 6.4M |
| DeepConvLSTM [35] | 91.4▼ | 67.2M | 71.6▼ | 80.4M | 85.5▼ | 68.3M | 62.0▼ | 50.4M |
| Bi-GRU-I [50] | 94.6▼ | 46.1M | 75.2▼ | 54.0M | 85.6▼ | 48.7M | 77.2▼ | 39.8M |
| RevAttNet [40] | 95.1▼ | 143.1M | 76.7▼ | 168.7M | 87.6▼ | 140.2M | 78.6▼ | 101.5M |
| IF-ConvTransformer [63] | 95.4 | 209.8M | 77.0▼ | 183.5M | 87.5▼ | 628.4M | 82.2 | 986.2M |
| T-ResNet [54, 12] | 95.3 | 123.2M | 76.5▼ | 145.5M | 87.3▼ | 125.8M | 80.9 | 96.9M |
| T-FCN [54, 12] | 95.8 | 68.9M | 76.9▼ | 80.6M | 86.7▼ | 76.1M | 76.2▼ | 65.8M |
| MILLET [11] | 94.7▼ | 111.6M | 81.4△ | 129.9M | 84.3▼ | 132.8M | 82.3 | 125.0M |
| DSN [55] | 95.4 | 270.8M | 79.8 | 320.0M | 86.4▼ | 265.7M | 71.8▼ | 192.1M |

| Model | KU-HAR | | PAMAP2 | | REALDISP | |
|---|---|---|---|---|---|---|
| | F1 | FLOPs | F1 | FLOPs | F1 | FLOPs |
| CALANet (**Ours**) | 97.5 | **29.6M** | 79.4 | **74.9M** | 98.2 | **56.7M** |
| CALA-GTSNet (**Ours**) | 96.1▼ | 12.2M | 76.3▼ | 23.9M | 95.4▼ | 43.3M |
| Shallow ConvNet [23] | 77.9▼ | 41.6M | 67.4▼ | 151.8M | 95.9▼ | 209.9M |
| RepHAR [49] | 93.4▼ | 74.4M | 73.0▼ | 131.9M | 94.7▼ | 72.7M |
| Res-GTSNet [37] | 94.5▼ | 15.3M | 76.2▼ | 28.6M | 94.9▼ | 18.0M |
| DeepConvLSTM [35] | 93.5▼ | 169.1M | 77.3▼ | 303.9M | 91.7▼ | 156.9M |
| Bi-GRU-I [50] | 94.9▼ | 108.0M | 71.0▼ | 194.1M | 96.1▼ | 111.3M |
| RevAttNet [40] | 97.7 | 335.3M | 79.7 | 573.5M | 98.5 | 282.1M |
| IF-ConvTransformer [63] | 96.4▼ | 491.7M | 80.1 | 1.7G | 97.4 | 3.0G |
| T-ResNet [54, 12] | 95.0▼ | 290.0M | 71.4▼ | 506.1M | 96.0▼ | 270.1M |
| T-FCN [54, 12] | 92.5▼ | 161.7M | 72.5▼ | 298.5M | 95.9▼ | 184.5M |
| MILLET [11] | 97.8 | 262.5M | 80.2 | 509.5M | 95.1▼ | 352.9M |
| DSN [55] | 97.1 | 634.8M | 68.8▼ | 1.08G | 97.5 | 532.7M |

**Baseline.** We compared CALANet with 11 baseline networks. To evaluate the efficiency of CALANet, we used Shallow ConvNet [23], RepHAR [49], and Res-GTSNet [37] as state-of-the-art models in real-time HAR. Meanwhile, we adopted four CNNs with recurrent layers or attention mechanisms, including DeepConvLSTM [35], Bi-GRU-I [50], RevAttNet [40], and IF-ConvTransformer [63], to verify the effectiveness of our all-layer aggregation. In addition, we used four networks, T-ResNet [54, 12], T-FCN [54, 12], MILLET [11], and DSN [55] that achieved substantial success in the time-series classification (TSC), which is more general-purpose than HAR. The details for the models and hyperparameters are described in Appendix G.

To evaluate the performance of CALANet, we used two metrics: F1-score and floating-point operations (FLOPs). Because the HAR datasets inherently involve a class imbalance, the F1-score has been commonly used as an alternative for accuracy. In particular, FLOPs have been widely used to describe how many operations a given model requires to run a single pattern. In addition, we investigate the change in performance according to $L$, as will be described in Section 4.3. Meanwhile, Res-GTSNet [37] can derive layer-specific structures suitable for a given computational budget, and the original paper adopted T-ResNet as an initial network. To improve the efficiency of CALANet further, we also designed the CALA-GTSNet by replacing T-ResNet with our CALANet.

## 4.2 Comparison results

Table 1 shows the results of comparing CALANet and the baseline networks. The experiments ran ten times, and the average values were recorded on all the datasets. In addition, we performed a paired

Table 2: Ablation study of CALANet on two datasets; LCTMs: Learnable channel-wise transformation matrices, SLAP: Scalable layer aggregation pool, ALA: All-layer aggregation

| Networks | L | KU-HAR | | PAMAP2 | |
| --- | --- | --- | --- | --- | --- |
| | | F1 | FLOPs | F1 | FLOPs |
| CALANet with LCTMs + SLAP | 9 | 97.5 | 29.7M | 79.4 | 74.9M |
| CALANet with LCTMs only | 4 | 93.8▼ | 60.0M | 73.1▼ | 113.3M |
| CALANet with ALA only | 4 | 95.0▼ | 577.9M | 72.8▼ | 1.7G |

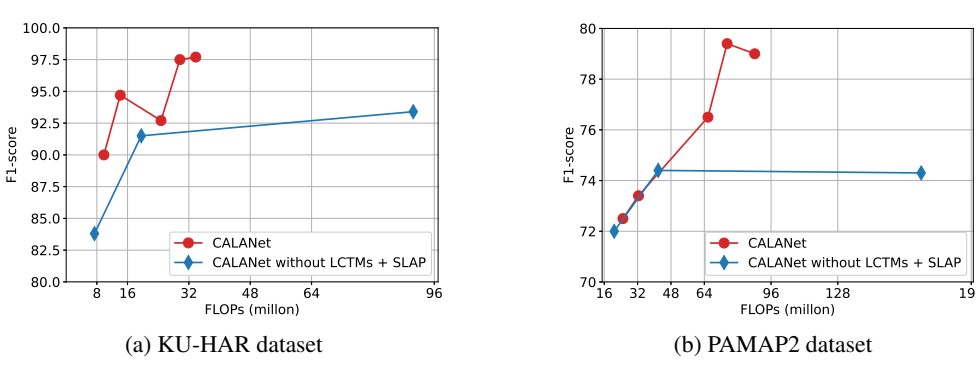

(a) KU-HAR dataset          (b) PAMAP2 dataset

Figure 3: Tradeoff between the FLOPs and F1-score.

$t$-test at the 95% significance level on each dataset. In Table 1, ▼/△ indicates that the compared network was significantly worse/better than CALALet regarding the F1-score.

**Comparison with real-time CNNs.** In Table 1, the F1-scores of CALANet were statistically superior to real-time HAR models on all datasets. In particular, CALANet has the lowest FLOPs compared to other real-time HAR models with standard convolution layers on seven datasets. Meanwhile, Res-GTSNet, with an efficient variant of the convolution, exhibited significantly low FLOPs. This variant can be easily integrated with our CALANet to reduce its FLOPs further. As shown in Table 1, CALA-GTSNet outperformed Res-GTSNet on 86% of the datasets. Also, CALA-GTSNet has lower FLOPs than Res-GTSNet on 71% of the datasets. As a result, CALANet and GTSNet can complement each other to improve the accuracy or reduce computations. These results demonstrated that our cheap all-layer aggregation can maintain a low computational cost.

**Comparison with accuracy-oriented networks.** We noted that real-time or efficient HAR models using wearable sensors process the input signals with short segmentation lengths for rapid response. If CNNs are sufficient to extract meaningful information from the short-term signals, unnecessary increases in inference time due to integration with recurrent layers or attention mechanisms can be avoided. In Table 1, CALANet outperformed two CNNs with recurrent layers, i.e., DeepConvLSTM and Bi-GRU-I, on all datasets. Compared with RevAttNet and IF-ConvTransformer, CALANet exhibited a comparable F1-score despite its significantly low FLOPs. These results indicate that CNNs are sufficient to model the temporal information for the real-time HAR dataset. Compared with TSC models, CALANet showed comparable performance despite its significantly low FLOPs. These results demonstrated that our cheap all-layer aggregation can significantly improve HAR accuracy while maintaining low FLOPs.

### 4.3 Ablation study

**The breakdown effect of CALANet.** We conducted an ablation study to investigate the effectiveness and efficiency of our CALANet. The key components of CALANet are LCTMs and SLAP. Therefore, we compared the performance of our CALANet with that of its two variants, which were obtained by removing each component. The first variant removes the SLAP described in Section 3.3. The second variant replaces the LCTMs with fully-connected layers that have the same number of units as the input size. Table 2 shows that our LCTMs substantially reduced the FLOPs for calculating all-layer aggregation without losing the F1-score. In addition, the SLAP enhanced the effectiveness of all-layer

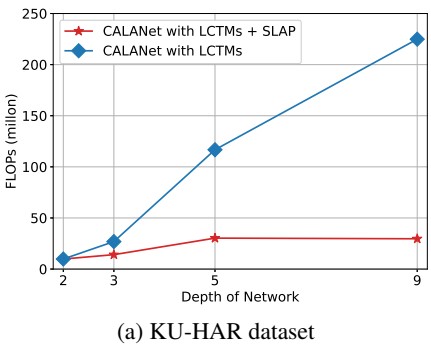

(a) KU-HAR dataset

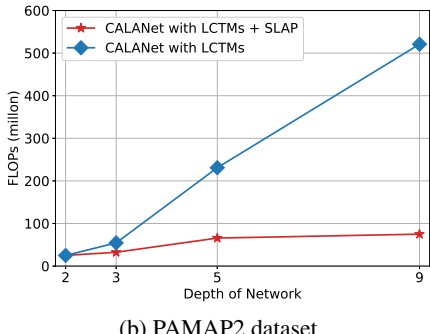

(b) PAMAP2 dataset

Figure 4: Comparison among two networks with regard to the impact of $L$ on the FLOPs.

aggregation even while reducing FLOPs. Especially, Figure 3 shows the tradeoff between the FLOPs and F1-score with varying numbers of layers in CALANet with/without LCTMs and SLAP. The tradeoff curves closer to the top-left are more efficient, with a higher F1-score per FLOPs. As shown in Figure 3, CALANet with LCTMs and SLAP achieved a higher F1-score in similar computational cost than one without LCTMs and SLAP.

**Effect of scalable layer aggregation pool.**
We investigated the layer aggregation pool at which the best F1-score of CALANet is achieved on seven datasets. Table 3 shows the change in F1-score of CALANet as the layer aggregation pool $L$ increased; herein, the best F1-score is indicated by the bold font on each dataset. As shown in Table 3, the layer aggregation pool and F1-score tend to be proportional. In Figure 4, CALANet with SLAP (red line) exhibited a negligible increase in FLOPs compared with CALANet only with LCTMs. As a result, SLAP allows CALANet to stack layers without significantly increasing FLOPs.

Table 3: F1-score of CALANet on different layer aggregation pool, i.e., network depth $L$

| Datasets | $L$ | | | | |
| --- | --- | --- | --- | --- | --- |
| | 2 | 3 | 5 | 9 | 17 |
| UCI-HAR | 93.2 | 93.9 | 94.8 | **96.1** | 95.7 |
| UniMiB-SHAR | 72.8 | 76.0 | 78.0 | **78.3** | 77.5 |
| DSADS | 87.2 | 84.5 | 86.0 | **90.0** | 89.4 |
| OPPORTUNITY | 78.9 | 80.2 | 79.0 | **81.6** | 80.3 |
| KU-HAR | 90.0 | 94.7 | 92.7 | 97.5 | **97.7** |
| PAMAP2 | 72.5 | 73.4 | 76.5 | **79.4** | 79.0 |
| REALDISP | 92.9 | 96.7 | 96.9 | **98.2** | 97.7 |

**Performance analysis on similar activities.** To verify the performance of CALANet, we investigated the confusion matrices (see Appendix H). Prior works [47, 24] suffered from activities that have similar vibrations in signal waveforms, such as "Sit" and "Talk-Sit," as described in Section 1. Compared with these works, our CALANet significantly improved the accuracy of those activities on the KU-HAR dataset. For other examples, these activities include ("rope jumping" and "waking") [20, 14, 48, 49] and ("knees bending crouching" and "reach heels backwards") [8]. On the other hand, our CALANet correctly classified "rope jumping" as "waking" compared to RepHAR that misclassified "rope jumping" as "walking" 20 times [49] on the PAMAP dataset. Compared to MG-WHAR [8] misclassified "knees bending crouching" by approximately 20% as "reach heels backwards", our CALANet misclassified "knees bending crouching" as "reach heels backwards" only two times on the REALDISP dataset.

**Applicability of CALA structure.** Our CALA structure can effectively be applied to existing CNNs if the following constraints are satisfied: (1) the layers of a network architecture should be calculated sequentially and independently; (2) the output of each layer should be able to be expressed as a (temporal length $\times$ channel size) matrix. To the best of our knowledge, most wearable sensor-based human activity recognition models can satisfy the above constraints. In Table 4, we applied our LCTMs and SLAP to SqueezeNet [22]. Specifically, the output of a squeeze convolution layer in each fire module is fed into LCTMs and connected to the last layer. As a result, our modification significantly improved the F1-score of SqueezeNet on 71% of all datasets while maintaining its FLOPs. In addition, we applied CALANet to the ECG heartbeat classification problem using the MIT-BIH arrhythmia dataset [16]. CALANet exhibited comparable performance with other networks

Table 4: Comparison results of SqueezeNet with/without CALA structure on seven datasets.

| Model | UCI-HAR | | UniMiB-SHAR | | DSADS | | OPPORTUNITY | |
| | F1 | FLOPs | F1 | FLOPs | F1 | FLOPs | F1 | FLOPs |
|---|---|---|---|---|---|---|---|---|
| SqueezeNet + CALA (**Ours**) | 92.4 | 8.2M | 75.8 | 9.5M | 87.3 | 11.4M | 68.4 | 12.3M |
| SqueezeNet | 92.1 | 10.4M | 74.9 | 12.4M | 84.7▼ | 13.7M | 59.7▼ | 13.2M |

| Model | KU-HAR | | PAMAP2 | | REALDISP | |
| | F1 | FLOPs | F1 | FLOPs | F1 | FLOPs |
|---|---|---|---|---|---|---|
| SqueezeNet + CALA (**Ours**) | 96.7 | 19.5M | 70.4 | 41.5M | 93.7 | 34.8M |
| SqueezeNet | 94.5▼ | 25.5M | 68.0▼ | 52.9M | 85.6▼ | 39.7M |

[13, 39] designed to process ECG signals (see Appendix I). This result shows that CALANet has promising applicability to other ML applications.

## 4.4 Real-Time Activity Prediction

To estimate the actual response time of our CALANet, we used the AMD Ryzen 7 5800X 8-Core Processor without the support of graphics processing units. Particularly, we compared the inference time of CALANet with Shallow ConvNet. Similar to the conventional real-time HAR studies, the measurements were repeated 1,000 times, and the minimum, maximum, and mean values were recorded. Table 5 shows the inference time of the two networks, where the

Table 5: Actual inference time of CALANet

| Model | Inference Time (ms / window) | | |
| | Min | Mean | Max |
|---|---|---|---|
| CALANet | 1.59ms | 2.25ms | 3.40ms |
| Shallow ConvNet | 1.57ms | 2.15ms | 3.48ms |

window length was set to 300 (3 *s*) to slide one instance at a time. CALANet exhibited a response time similar to Shallow ConvNet, even though its depth is nine times deeper than that of Shallow ConvNet. Consequently, these measurements show that our model is sufficient to meet the real-time requirements.

## 5 Conclusion

In this article, we proposed an effective neural network called CALANet for real-time HAR from wearable sensors. In particular, our CALANet has an all-layer aggregation structure that can aggregate features for all layers based on the learnable channel-wise transformation matrix and scalable layer aggregation pool. As a result, CALANet improved HAR accuracy while maintaining the efficiency of existing real-time HAR models. In addition, we proved that the computational cost of CALANet is equivalent to that of shallow CNNs. Our experiments demonstrated that CALANet could achieve state-of-the-art performance on the HAR datasets under low latency.

Future studies can be conducted to overcome the limitations of the proposed method. CALANet does not consider the various computational budgets that can be changed according to the specific devices and the runtime optimizations of actual devices, such as memory access costs and parallel computations. For example, future studies may further improve CALANet by introducing a new operator designed to match the target device.

**Acknowledgement.** This research was supported in part by the Institute of Information & Communications Technology Planning & Evaluation (IITP) grant funded by the Korean Government (MSIT) (2021-0-01341, Artificial Intelligence Graduate School Program (Chung-Ang University)), in part by the Institute of Information & Communications Technology Planning & Evaluation (IITP) grant funded by the Korean Government (MSIT) (2021-0-00766, Development of Integrated Development Framework that supports Automatic Neural Network Generation and Deployment optimized for Runtime Environment), and in part by the National Research Foundation of Korea (NRF) grant funded by the Korea government (MSIT) (No. 2023R1A2C1006745).

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

# Appendix

## A    Proof of Proposition 1

Given the original signals $X^{(0)} \in \mathbb{R}^{\mathbb{T},\mathbb{M}}$, the first convolution layer generates new feature signals $X^{(1)} \in \mathbb{R}^{\mathbb{T},N^1}$, resulting in a time complexity of $\mathcal{O}(\mathbb{T}\mathbb{M}N^{(1)}D_k^{(1)})$. Suppose that the $l$-th intermediate convolution layer includes a kernel $K^{(l)} \in \mathbb{R}^{D_k^{(l)},N^{(l-1)},N^{(l)}}$ and $N^{(l)}$ is a positive-integer multiple of $N^{(l-1)}$, that is, $N^{(l)} = c^{(l)}N^{(l-1)}$. Generally, $c \geq 1$ and the pooling layer adjusts the temporal resolution $T$ to $T/c$ if $c \geq 2$. Therefore, its time complexity is $\mathcal{O}(T^{(l)}D_k^{(l)}M^{(l)}N^{(l)})) = \mathcal{O}(T^{(l-1)}D_k^{(l)}(N^{(l-1)})^2)$. Given the stack of $L$ convolution layers, their time complexity can be simplified as follows:

$$\underbrace{\mathcal{O}(\mathbb{T}\mathbb{D}_k\mathbb{M}\mathbb{N})}_{Part\ 1} + \mathcal{O}(\mathbb{T}\mathbb{D}_k\mathbb{N}^2 L), \tag{10}$$

where $\mathbb{N}$ and $\mathbb{D}_k$ are the average of the number of output features and the kernel sizes across the layers, respectively. If $\mathbb{M} \leq \mathbb{N}(L-1)$, the time complexity of convolution layers becomes $\mathcal{O}(\mathbb{T}\mathbb{D}_k\mathbb{N}^2 L)$.

## B    Proof of Lemma 1

Consistent with Proposition 1, the time complexity of calculating the features is $\mathcal{O}(\mathbb{T}\mathbb{D}_k\mathbb{N}^2 L)$. Since the number of output channels across the standard convolution layers is $\mathbb{N}$, the time complexity of calculating LCTMs for all layers is $\mathcal{O}(\mathbb{T}\mathbb{N}^2 L)$. Given $L \times \mathbb{N}$ aggregated features and $V$ activities, the Softmax layer predicts an activity with the time complexity of $\mathcal{O}(L\mathbb{N}V)$. Consequently, the time complexity of our CALANet is formalized as:

$$\underbrace{\mathcal{O}(\mathbb{T}\mathbb{D}_k\mathbb{N}^2 L)}_{Part\ 2} + \underbrace{\mathcal{O}(\mathbb{T}\mathbb{N}^2 L)}_{Part\ 3} + \underbrace{\mathcal{O}(L\mathbb{N}V)}_{Part\ 4}. \tag{11}$$

Because $V$ commonly is less than $\mathbb{T} \times \mathbb{D}_k \times \mathbb{N}$, Eq. (11) can be rewritten as $\mathcal{O}(\mathbb{T}\mathbb{D}_k\mathbb{N}^2 L)$.

## C    Proof of Lemma 2

Given an input $X \in \mathbb{R}^{T,M}$, $M$ channels are evenly divided into $L$ channel groups. More precisely, for each channel group, the convolution layer generates $\lfloor N/L \rfloor$ output channels, resulting in the time complexity of $\mathcal{O}(TD_k(MN/L^2))$. Since each convolution layer generates the output $Y \in \mathbb{R}^{T,N}$ across $L$ channel groups, the time complexity of each layer is $\mathcal{O}(TD_k(MN/L))$. Because our CALANet includes $L$ standard convolution layers to calculate the local temporal correlations, part 2 of Eq. (11) is reduced to $\mathcal{O}(\mathbb{T}\mathbb{D}_k\mathbb{N}^2)$, consistent with Proposition 1.

## D    Proof of Lemma 3

Given an input $X \in \mathbb{R}^{T,M}$, LCTMs at each layer generate a vector with $\lfloor \mathbb{N}/L \rfloor$ elements, resulting in the time complexity of $\mathcal{O}(TM(\mathbb{N}/L))$. As shown in Figure 2, the number of output channels for each layer is fixed as $\mathbb{N}$ across layers. Therefore, the time complexity of the all-layer aggregation is $\mathcal{O}(T(\mathbb{N}^2/L))$. Similar to Lemma 2, part 3 of Eq. (11) is reduced to $\mathcal{O}(\mathbb{T}\mathbb{N}^2)$.

## E    Proof of Theorem 1

Consistent with Lemma 2 and Lemma 3, part 2 and part 3 of Eq. (11) is simplified as:

$$\mathcal{O}(\mathbb{T}\mathbb{D}_k\mathbb{N}^2) + \mathcal{O}(\mathbb{T}\mathbb{N}^2) \tag{12}$$

In addition, we fixed the number of features fed into the Softmax layer to $\mathbb{N}$. Therefore, part 4 of Eq. (11) is reduced to $\mathcal{O}(\mathbb{N}V)$. Because $V$ commonly is less than $\mathbb{T} \times \mathbb{D}_k \times \mathbb{N}$, Eq. (6) is reduced $\mathcal{O}(\mathbb{T}\mathbb{D}_k\mathbb{N}^2)$.

# F    Details of datasets

The benchmark datasets used in our experiment follow the following setup:

- The **UCI-HAR** dataset [1] was recorded at a sampling frequency of 50 Hz. Precisely, 30 subjects performed six basic activities (e.g., walking, upstairs, sitting) using accelerometers and gyroscopes embedded in Android smartphones. As the authors recommended, each segment's length is set to 128, and 70% and 30% of the dataset were used as the training and test sets, respectively.

- The **UniMiB-SHAR** dataset [30] was recorded at a sampling frequency of 50 Hz, where the length of each segment is 151. The 30 subjects performed 17 activities, including nine activities of daily living (e.g., walking and standing) and eight fall activities (e.g., forward and syncope), using an accelerometer in Android smartphones. Precisely, 70% and 30% of the dataset were used as the training and test sets, respectively.

- The **DSADS** dataset [3] was recorded at a sampling frequency of 25 Hz, where the length of each segment is 125. Eight subjects performed 19 daily and sports activities (e.g., exercising on a stepper and rowing) using accelerometers, gyroscopes, and magnetometers embedded in five MTx trackers. More precisely, the MTx units measured the sensor signals on the torso, right arm, left arm, right leg, and left leg. We split the dataset into 80% training and 20% test sets based on the subject's identification (ID).

- The **OPPORTUNITY** dataset [6] was recorded at a sampling frequency of 30 Hz in a sensor-rich environment with wearable, object, and ambient sensors, where the length of each segment is 90. We only considered wearable sensors for real-time HAR, including accelerometers and inertial measurement units (IMUs); the number of input channels is 113 in our experiments. The four subjects performed 17 activities, including complicated activities such as "drink from cup" and "open door," categorized into ADL 1-5 and Drill. We used the ADL 5 data for subject 1; the Drill data for subject 2; the ADL 1 and 4 data for subject 3; and the ADL 4 data for subject 4 as the test set [32]. The remaining data were used as the training set.

- The **KU-HAR** dataset [47] was recorded at a sampling frequency of 100 Hz, where the length of each segment is 300. Precisely, 90 subjects performed 18 daily activities (e.g., talking with hand movements and picking up an abject) using accelerometers and gyroscopes embedded in smartphones. The 80% and 20% of the dataset were used as the training and test sets, respectively.

- The **PAMAP2** dataset [46] was recorded at 100 and 9 Hz sampling frequencies for IMUs and a heart rate monitor, respectively. Precisely, nine subjects performed 18 activities, including basic activities (e.g., sitting and running) and complicated activities (e.g., watching TV and folding laundry), where the length of each segment is 512. We used the data aggregated from subjects 102 and 106 as the test set. In addition, we added 30% of the data for "watching TV", "car driving," and "playing soccer" into the test set because they were performed by only one subject. The remaining data were used as the training set.

- The **REALDISP** dataset [2] was recorded at a sampling frequency of 50 Hz from nine IMUs, where the length of each segment is 250. More precisely, 17 subjects performed 33 fitness activities (e.g., lateral bend arm up and upper trunk and lower body opposite twist) using accelerometers, gyroscopes, and magnetometers. In addition, the authors provided orientation estimates in quaternion format. We split the dataset into 70% training and 30% test sets based on subject ID.

# G    Details of baselines

We summarize the models used in our experiments, as follows:

- **CNNs for real-time HAR.** We adopted three CNNs as state-of-the-art models in real-time HAR. Shallow ConvNet [23] comprises a single convolution layer and two fully-connected layers, where the basic statistical features to encode global temporal information are concatenated with outputs of the convolution layer. RepHAR [49] comprises three

convolution layers with re-parameterization and a Softmax layer. Res-GTSNet [37] contains nine grouped temporal shift module layers and a Softmax layer.

- **CNNs with recurrent layers or attention mechanisms.** We adopted two CNNs with recurrent layers and two CNNs with attention mechanisms to verify the effectiveness of our all-layer aggregation. DeepConvLSTM [35] contains four convolution layers and two recurrent LSTM layers with a Softmax classifier. Bi-GRU-I [50] is composed of two bi-directional GRU layers, three inception layers, and a Softmax layer. Compared with DeepConvLSTM, its architecture has a reverse order of convolution and recurrent layers. RevAttNet [40] contains six convolutional layers, two recurrent layers, and two reverse attention modules, including LSTM, deconvolution, and multi-head attention. IF-ConvTransformer [63] is composed of IMU fusion blocks, four convolutional layers, and two self-attention layers, where the IMU fusion blocks are used to fuse the features from multiple sensor modalities based on sensor-wise convolutional layers.

- **Time-Series Classification (TSC) models.** In addition, we adopted two CNNs that achieved substantial success in the TSC, which is more general-purpose than HAR. Specifically, T-ResNet [54, 12] consists of three residual blocks, each comprising three convolution layers with a residual connection, and a Softmax layer. T-FCN [54, 12] consists of three convolution layers and a Softmax layer. Compared with the real-time models, it has more output channels. MILLET [11] comprises five identical networks with conjunctive pooling layers, where each network contains six convolutional layers with multiple kernel sizes. DSN [55] contains three sparse CNN module, each of which includes a dynamic sparse convolution and point-wise convolution.

For fairness, we re-implemented all the baseline networks in PyTorch [38] and excluded sophisticated tricks of each original setting, such as a gradient clipping [31] and learning rate warmup [18]. Precisely, they were trained for 300 epochs with a batch size of 128 using a 2080Ti graphics-processing unit. We used the Adam optimizer [26] with $\beta_1 = 0.9$, $\beta_2 = 0.999$ and $\epsilon = 10^{-8}$, where the learning rate and weight decay were set to 0.0005. For the CALALet, we set $\mathbb{D}_k$, $\mathbb{N}$, and $L$ to 5, 128, and 9, respectively. On the OPPORTUNITY and REALDISP datasets with many input channels (113 and 117), we doubled the number of filters for the first convolution of CALANet. The hyper-parameters of the existing networks are set according to the values recommended in the original papers.

## H  Confusion matrices

Figures 5–7 show the confusion matrices of CALANet on three datasets. Their diagonal elements represent the number of instances correctly classified to the related activities, with the color darkening as the number grows.

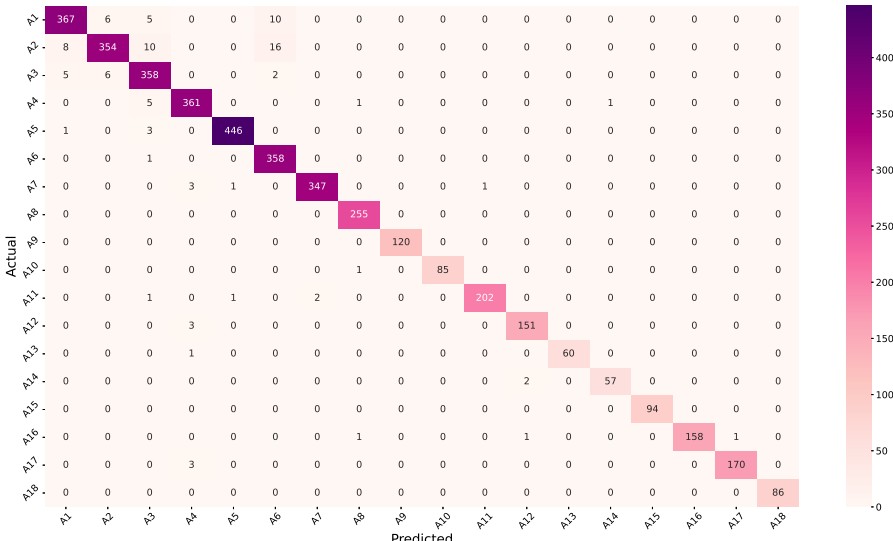

Figure 5: Confusion matrix of CALANet on the KU-HAR dataset. A1, stand; A2, sit; A3, talk-sit; A4, talk-stand; A5, stand-sit; A6, lay; A7, lay-stand; A8, pick; A9, jump; A10, push-up; A11, sit-up; A12, walk; A13, walk-backward; A14, walk-circle; A15, run; A16, stair-up; A17, stair-down; A18, table-tennis.

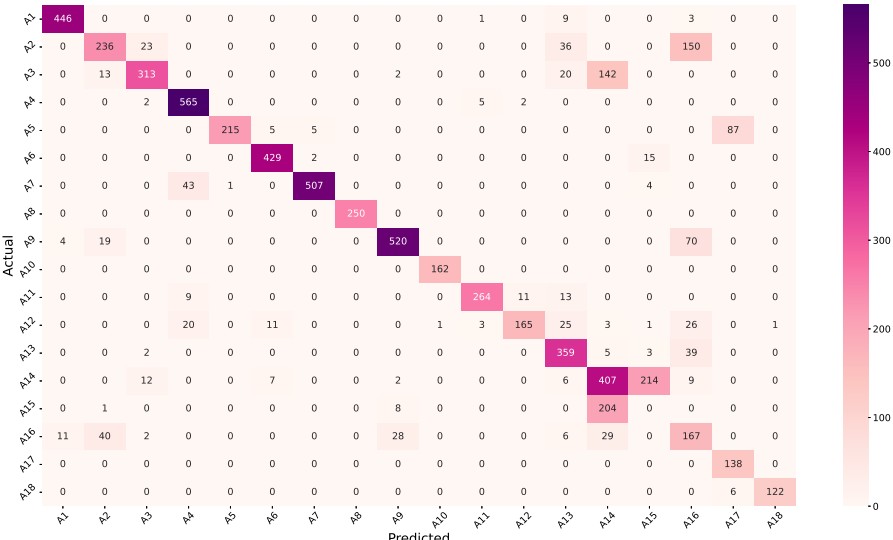

Figure 6: Confusion matrix of CALANet on the PAMAP2 dataset. A1, lying; A2, sitting; A3, standing; A4, walking; A5, running; A6, cycling; A7, Nordic-walking; A8, watching-TV; A9, computer-work; A10, car-driving; A11, ascending-stairs; A12, descending-stairs; A13, vacuum-cleaning; A14, ironing; A15, folding-laundry; A16, house-cleaning; A17, playing-soccer; A18, rope-jumping.

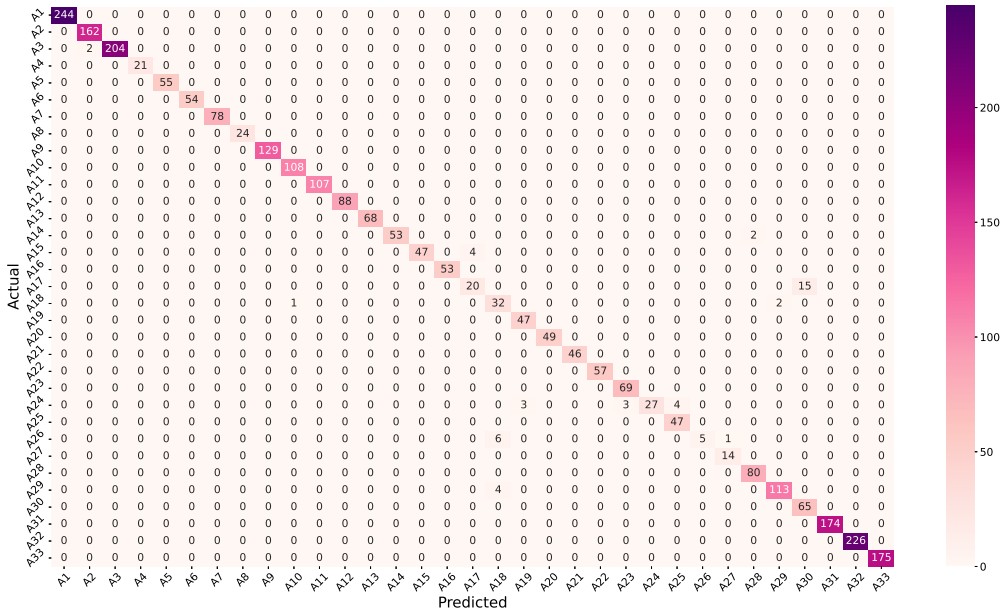

Figure 7: Confusion matrix of CALANet on the REALDISP dataset. A1, walking; A2, jogging; A3, running; A4, jump-up; A5, jump-front-back; A6, jump-sideways; A7, jump-leg/arms-open/closed; A8, jump-rope; A9, trunk-twist-arms; A10, trunk-twist-elbows; A11, waist-bends-forward; A12, waist-rotation; A13, waist-bends; A14, reach-heels-backwards; A15, lateral-bend; A16, lateral-bend-arm-up; A17, repetitive-forward-stretching; A18, upper-trunk-and-lower-body-opposite-twist; A19, arms-lateral-elevation; A20, arms-frontal-elevation; A21, frontal-hand-claps; A22, arms-frontal-crossing; A23, shoulders-high-amplitude-rotation; A24, shoulders-low-amplitude-rotation; A25, arms-inner-rotation; A26, knees-alternatively-breast; A27, heels-alternatively-backside; A28, knees-bending-crouching; A29, knees-alternatively-bend-forward; A30, rotation-on-the-knees; A31, rowing; A32, elliptical-bike; A33, cycling.

## I ECG heartbeat classification

We applied CALANet to the ECG heartbeat classification problem using the MIT-BIH arrhythmia dataset [16], which includes 24-hour ambulatory ECG recordings collected from inpatients and outpatients at Boston's Beth Israel Hospital. The dataset has 21,892 heartbeats, each with a signal length of 187. In Table 6, CALANet exhibited comparable performance with other networks designed to process ECG signals. This result shows that CALANet has promising applicability to other ML applications.

Table 6: Comparison results on the MIT-BIH arrhythmia dataset.

| Model | Average Accuracy |
| --- | --- |
| CALANet | 98.2 |
| Pham et al. [39] | 98.5 |
| Ganguly et al. [13] | 97.3 |

