# OpenReview forum: "CALANet: Cheap All-Layer Aggregation for Human Activity Recognition"
_NeurIPS.cc/2024/Conference — NeurIPS 2024 poster_

### Official Review · Reviewer_2n6z · 2024-06-27

**Soundness:** 2
**Presentation:** 3
**Contribution:** 2
**Rating:** 4
**Confidence:** 3

**Summary:**

This paper designs CALANet, a cheap all-layer aggregation network designed for real-time sensor-based  HAR. The main objective of CALANet is to improve the accuracy of HAR while maintaining a low computational costs on edge devices for real-time applications. The authors argue that existing CNN models for HAR often suffer from limited accuracy because they only use features from the last layer of the network. In contrast, CALANet allows the classifier to aggregate features from all layers. The authors theoretically prove that the computational cost of CALANet is equivalent to that of conventional CNNs. The authors utilize 7 datasets to demonstrate the effectiveness of CALANet. The results show that CALANet outperforms seven state-of-the-art methods, achieving superior performance on 7 datasets.

**Strengths:**

[1] The topic of sensor-based HAR is interesting and important, aligning with the scope of the ML community.

[2] I appreciate authors for providing theoretical proofs, for the complexity and others.

[3] 7 datasets are used in evaluation, providing sufficient evaluation results with detailed analysis. Both efficiency and effectiveness are evaluated.

[4] It is good to see future studies are also discussed.

**Weaknesses:**

-  One of my concerns is the baselines. Why choose these models as baselines? I believe there are many more advanced models in sensor-based HAR. I suggest the authors go through some papers in AAAI, IMWUT, Sensys, Mobicom, where most HAR papers are published. Otherwise, current improvement might not justify the advantage of CALANet. Also, there are more advanced time series classification models from NIPS, ICML, etc.
-  The goal of this study is improving the accuracy of HAR with the computational costs during the inference phase as the constraint. Therefore, the fundamental thing is the accuracy. However, why CNN become the choice for this study? While authors mention that CNN is popular for HAR recently in Line 27, this argument does not support the study of accelerating CNN for HAR in this paper.
-  Also curious about the dataset partition, some are 7:3 and others are 8:2.
-  More references added in the content will improve the manuscript further, e.g., from line 51 to 55, references can be added to support the argument of the “” straightforward approach” and the “”computational costs”.
-  In related work section, it seems the discussion of accelerating CNN or neural network is missing, which is a popular topic and has been widely studied.
-  Writing: Writing can be further improved. For example, in Table 1, authors can put their own models at the same location (e.g., at the bottom). I believe this will lower the cognitive load of readers. In line 47, “under real-time response”.
-  I suggest the authors to re-design Fig 2 for better readability. Fig 2 takes a lot of spaces but it is challenging to understand why the proposed model can utilize all features with low computational costs.

**Questions:**

Please see above.

**Limitations:**

Please see above.

---

> ### Author Rebuttal · Authors · 2024-08-07
>
> We are grateful for the constructive comments of the reviewer. Below, we provide specific answers and explanations regarding those comments.
>
> **(W1) One of my concerns is the baselines. Why choose these models as baselines? I believe there are many more advanced models in sensor-based HAR. I suggest the authors go through some papers in AAAI, IMWUT, Sensys, Mobicom, where most HAR papers are published. Otherwise, current improvement might not justify the advantage of CALANet. Also, there are more advanced time series classification models from NIPS, ICML, etc.**
>
> Thank you for your constructive comments. Although there are many advanced models in sensor-based HAR, unfortunately, we could not find the advanced models in “real-time HAR.”
> For example, generative models or seme-supervised HAR studies are difficult to compare fairly with our CALANet due to different evaluation scenarios.
> Instead, we experimented with two models in time series classification. Please check the attached .pdf file in “Global(Author) Rebuttal”. Table 2 shows the comparison results for CALANet, MILLET, and DSN. MILLET is a TSC framework designed to provide inherent interpretability. Meanwhile, DSN was proposed so that the temporal receptive field is trained sparsely and selectively.
> As shown in Table 2, CALANet exhibited comparable performance despite its significantly low FLOPs.
>
> **(W2) The goal of this study is improving the accuracy of HAR with the computational costs during the inference phase as the constraint. Therefore, the fundamental thing is the accuracy. However, why CNN become the choice for this study? While authors mention that CNN is popular for HAR recently in Line 27, this argument does not support the study of accelerating CNN for HAR in this paper.**
>
> Thank you for your constructive comments. Popular architectures in sensor-based human activity recognition (HAR) literature include CNNs, RNNs, and Transformers.
> However, our goal is to provide accurate feedback while satisfying real-time constraints. In real-time HAR, RNNs have some weaknesses, including poor parallelization and the lack of hardware accelerators compared to CNNs.
> In addition, Transformer needs to calculate the relationship between timestamp of which each is meaningless. Specifically, its search space is larger than CNNs, resulting in slow training time.
> Although the fundamental thing in this study is accuracy, other architectures find it difficult to satisfy the real-time constraint. Therefore, we adopted CNN. We apologize for the confusing sentences. To avoid these confusions, we will modify the corresponding sentences.
>
> **(W3) Also curious about the dataset partition, some are 7:3 and others are 8:2.**
>
> Thank you for your constructive comments. We used the values recommended from original paper or followed the setting that the prior studies adopted.
>
> **(W4) More references added in the content will improve the manuscript further, e.g., from line 51 to 55, references can be added to support the argument of the “” straightforward approach” and the “”computational costs”.**
>
> Thank you for your constructive comments. We will add not only two references [1], [2] but also more references to improve the manuscript further.
>
> **(W5) In related work section, it seems the discussion of accelerating CNN or neural network is missing, which is a popular topic and has been widely studied.**
>
> Thank you for your constructive comments. We will add some sentences. For example, furthermore, model compression technology is commonly used to accelerate inference time further on wearable or mobile devices. Especially recent studies preferred quantization methods over other compression methods, such as pruning [3]. This postprocessing can further accelerate and optimize CALANet.
>
> **(W6) Writing: Writing can be further improved. For example, in Table 1, authors can put their own models at the same location (e.g., at the bottom). I believe this will lower the cognitive load of readers. In line 47, “under real-time response”.**
>
> The entire sentence will be reviewed and modified to improve readability.
>
> **(W7) I suggest the authors to re-design Fig 2 for better readability. Fig 2 takes a lot of spaces but it is challenging to understand why the proposed model can utilize all features with low computational costs.**
>
> Thank you for your constructive comments. Although the figure still takes up a lot of space and cannot be attached, we hope it can be shown after further development.
>
> > [1] Lee, Chen-Yu, et al. "Deeply-supervised nets." Artificial intelligence and statistics. Pmlr, 2015.
>
> > [2] Yu, Fisher, et al. "Deep layer aggregation." Proceedings of the IEEE conference on computer vision and pattern recognition. 2018.
>
> > [3] Kuzmin, Andrey, et al. "Pruning vs quantization: which is better?." Advances in neural information processing systems 36 (2024).

---

> > ### Comment · Reviewer_2n6z · 2024-08-11
> >
> > I appreciate the detailed feedback from the authors, which addressed some concerns.
> >
> > One thing about "real-time HAR" is that some studies might not exactly highlight "real-time" in the title or abstract, however, they are still highly efficient. Also, in (w2), there are multiple claims about the efficiency of RNN and transformers. I suggest that the authors provide better intuition or motivation for choosing CNN. Also, some studies or tutorials focusing on the computational complexities of different model architectures might also be helpful.

---

> > > ### Author Response · Authors · 2024-08-14
> > >
> > > Thank you for your constructive comments.
> > >
> > > In this study, one of our primary missions is to achieve an accurate HAR without exceeding the model complexity of existing NNs for HAR.
> > > Thus, increasing the complexity of the model is not an option in our study because a larger model typically yields better accuracy with increasing inference time.
> > > As a result, we had to choose CNNs because CNNs are much lighter than RNNs or transformers, and CNNs for HAR already exist.
> > >
> > > Please see the following detailed explanation:
> > >
> > > Sensor-based HAR can be defined as a multivariate time series classification task.
> > > To solve this problem, the classifier requires both local (via CNNs) and global (via RNNs or transformers) temporal representations [1].
> > > Specifically, the locality of CNNs improves accuracy due to their translational invariance concerning the precise location of activity within a segment of time-series data [2].
> > > On the other hand, RNNs or transformers have an advantage for global feature extraction because they can model long-term dependencies.
> > > In this regard, many studies have attempted to integrate RNNs or transformers into CNNs [1,3-6], which has increased both accuracy and inference times.
> > > The increase in inference time is primarily because of the lack of device-level optimizations compared with CNNs [7].
> > >
> > > We noted that real-time or efficient HAR models using wearable sensors processes the input signals with short segmentation lengths for rapid response.
> > > If CNNs are sufficient to extract meaningful information from the short-term signals, unnecessary increase of inference time due to integration with RNNs or transformers can be avoided.
> > > In Table 1 of the manuscript, CALANet outperformed two CNNs with RNNs, i.e., Bi-GRU-I [5] and DeepConvLSTM [6], on all datasets.
> > > In addition, we compare CALANet with RevAttNet [3] and IF-ConvTransformer [4], hybridizations of CNNs and transformers.
> > > In the below Table, CALANet exhibited comparable performance despite its significantly low FLOPs.
> > > These results indicate that CNNs are sufficient to model the temporal information for the real-time HAR dataset.
> > >
> > > |(F1-score / FLOPs)|UCI-HAR|UniMiB-SHAR|DSADS|OPPORTUNITY| KU-HAR|PAMAP2|
> > > |:---|:---|:---|:---|:---|:---|:---|
> > > |CALANet|96.1 / 7.6M|78.3 / 8.8M|90.0 / 8.5M|81.6 / 19.3M|97.5 / 29.6M|79.4 / 74.9M|98.2 / 56.7M|
> > > |RevAttNet|95.1 / 143.1M|76.7 / 168.7M|87.6 / 140.2M|78.6 / 101.5M|97.7 / 335.3M|79.7 / 573.5M|98.5 / 282.1M|
> > > |IF-ConvTransformer|95.4 / 209.8M|77.0 / 183.5M|87.5 / 628.4M|82.2 / 986.2M|96.4 / 491.7M|80.1 / 1.7G|97.4 / 3.0G|
> > >
> > >
> > > > [1] Zhao, Bowen, et al. "Rethinking attention mechanism in time series classification." Information Sciences 627 (2023): 97-114.
> > >
> > > > [2] Hammerla, Nils Y., Shane Halloran, and Thomas Plötz. "Deep, convolutional, and recurrent models for human activity recognition using wearables." Proceedings of the Twenty-Fifth International Joint Conference on Artificial Intelligence. 2016.
> > >
> > > > [3] Pramanik, Rishav, Ritodeep Sikdar, and Ram Sarkar. "Transformer-based deep reverse attention network for multi-sensory human activity recognition." Engineering Applications of Artificial Intelligence 122 (2023): 106150.
> > >
> > > > [4] Zhang, Ye, et al. "IF-ConvTransformer: A framework for human activity recognition using IMU fusion and ConvTransformer." Proceedings of the ACM on Interactive, Mobile, Wearable and Ubiquitous Technologies 6.2 (2022): 1-26.
> > >
> > > > [5] Tong, Lina, et al. "A novel deep learning Bi-GRU-I model for real-time human activity recognition using inertial sensors." IEEE Sensors Journal 22.6 (2022): 6164-6174.
> > >
> > > > [6] Ordóñez, Francisco Javier, and Daniel Roggen. "Deep convolutional and lstm recurrent neural networks for multimodal wearable activity recognition." Sensors 16.1 (2016): 115.
> > >
> > > > [7] Mehta, Sachin, and Mohammad Rastegari. "MobileViT: Light-weight, General-purpose, and Mobile-friendly Vision Transformer." International Conference on Learning Representations. 2022.

---

### Official Review · Reviewer_VmYr · 2024-07-08

**Soundness:** 3
**Presentation:** 3
**Contribution:** 3
**Rating:** 6
**Confidence:** 4

**Summary:**

The CALANet describes a technique to aggregate the features from all the neural network layers for human activity recognition (HAR).
Because HAR is a common application for wearable devices, the model needs to be lightweight to be deployed on the edge for example on an Apple Watch.
Existing studies are often limited by shallow networks. The activity prediction is done on the final layer without using the features from previous layers,
which may contain key information.  Working with computational constraints, CALANet proposes to aggregate features from all the layers to improve the model performance.

The CALANet consists of two modifications: (1) a channel-wide transformation matrix to condense features from each layer (2) A ShuffleNet-like convolution to
aggregate all the layer features with computational efficiency. The authors also presented proof results on why CALANet is within the same computational complexity as a shallow network. The empirical results on 7 benchmarks against the state-of-the-art models supported the authors' claim.

**Strengths:**

1. This paper presented both theoretical guarantees and empirical evidence to validate the computational efficiency of the proposed model.
2. It was cool to see the authors start the paper with an empirical observation on how the features at different layers might differ to motivate the work.
3. CALANet addresses an important question for efficient mobile computing in the HAR space.
4. The manuscript reads well.

**Weaknesses:**

1. The notations of the proofs can be better clarified so that the readers don't need to refer to the appendix. For example, D is not explained in eq. 2 during its first occurrence.
2. This is a limitation of the field of HAR in general. Existing benchmarks are small usually with the number of participants under 100. It is fairly easy to overfit on the test set. I can see that you are just doing a simple train/test split. I know that some of the older benchmark recommends this but would be interesting to see cross-fold validation results. It is probably ok if you don't have the time to do this. Furthermore, could you clarify how you selected the hyper-parameters for all the models? Again, I suspect that the results reported could overfit your current test set given your evaluation framework.
3. In your ablation study you are trying to show the effectiveness of your layer aggregation technique (Table 2), there are several changing variables including network depths and FLOPs in addition to the network tricks you introduced. I don't think we can conclude that the layer aggregation trick worked. To test this properly, we should do the ablation studies using the same network L and probably similar FLOPs across different baselines.

**Questions:**

1. L60: what do you mean by temporal resolution T?
2. The majority if not all of the benchmark datasets you used are lab-based, which is really not a realistic assessment. Could consider adding one of the free-living datasets to the baselines if time allows. But you don't have to do this.
    1. Logacjov, Aleksej, et al. "HARTH: a human activity recognition dataset for machine learning." Sensors 21.23 (2021): 7853.
    2. Chan, Shing, et al. "CAPTURE-24: A large dataset of wrist-worn activity tracker data collected in the wild for human activity recognition." arXiv preprint arXiv:2402.19229 (2024).

---

> ### Author Rebuttal · Authors · 2024-08-07
>
> We are grateful for the constructive comments of the reviewer. Below, we provide specific answers and explanations regarding those comments.
>
> **(W1) The notations of the proofs can be better clarified so that the readers don't need to refer to the appendix. For example, $D$ is not explained in eq. 2 during its first occurrence.**
>
> **(Q1) L60: what do you mean by temporal resolution $T$?**
>
> Thank you for your constructive comments. We will modify or add some sentences that help understanding of notation, including a kernel size $D_k$ and a length of sequence $T$.
>
> **(W2) This is a limitation of the field of HAR in general. Existing benchmarks are small usually with the number of participants under 100. It is fairly easy to overfit on the test set. I can see that you are just doing a simple train/test split. I know that some of the older benchmark recommends this but would be interesting to see cross-fold validation results. It is probably ok if you don't have the time to do this. Furthermore, could you clarify how you selected the hyper-parameters for all the models? Again, I suspect that the results reported could overfit your current test set given your evaluation framework.**
>
> **(Q2) The majority if not all of the benchmark datasets you used are lab-based, which is really not a realistic assessment. Could consider adding one of the free-living datasets to the baselines if time allows. But you don't have to do this.**
>
> Thank you for your constructive comments. We agree that the recommendations of the older benchmark may cause an overfitting on the test set. We are conducting additional experiments and the results of CALANet's 5-fold cross-validation for KU-HAR dataset are as follows:
>
> > KU-HAR dataset (5-fold)
>
> > F1-score: 92.1  95.7  94.7  97.3  94.1
>
> Also, we almost used the hyper-parameters recommended from original paper except for epochs, batch size, and optimizer.
>
> **(W3) In your ablation study you are trying to show the effectiveness of your layer aggregation technique (Table 2), there are several changing variables including network depths and FLOPs in addition to the network tricks you introduced. I don't think we can conclude that the layer aggregation trick worked. To test this properly, we should do the ablation studies using the same network L and probably similar FLOPs across different baselines.**
>
> Thank you for your constructive comments. Please check the attached .pdf file in “Global(Author) Rebuttal”. In Table 1 we compared two networks with similar FLOPs and with/without our cheap all-layer aggregation. Despite applying the layer aggregation, its FLOPs were slightly reduced while improving F1-score. Especially, Figure 1 shows the tradeoff between the FLOPs and F1-score with varying numbers of layers in CALANet with/without LCTMs and SLAP. The tradeoff curves closer to the top-left are more efficient, with a higher F1-score per FLOPs. As a result, we can conclude that the layer aggregation trick worked.

---

> > ### Comment · Reviewer_VmYr · 2024-08-08
> >
> > Thank you very much for addressing all of my concerns.
> >
> > I can't upgrade my current rating from 6 to 7 because of the limited relevance of this manuscript because model architecture proposed are mostly applicable for bio-signals rather than time series in general.  However, a rating of 7 would require moderate impact on several domains. Perhaps, the ubiquitous computing community will find this work more relevant.

---

> ### Author Response · Authors · 2024-08-08
>
> Thank you for your recommendation and encouragement.
>
> Human Activity Recognition is a research topic of interest to the NeurIPS conference from the past to the present. Below are representative works presented at the NeurIPS conference.
>
> * DelPreto, Joseph, et al. "ActionSense: A multimodal dataset and recording framework for human activities using wearable sensors in a kitchen environment." Advances in Neural Information Processing Systems 35 (2022).
>
> * Cheng, Ricson, Ziyan Wang, and Katerina Fragkiadaki. "Geometry-aware recurrent neural networks for active visual recognition." Advances in Neural Information Processing Systems 31 (2018).
>
> * Mahdaviani, Maryam, and Tanzeem Choudhury. "Fast and scalable training of semi-supervised CRFs with application to activity recognition." Advances in Neural Information Processing Systems 20 (2007).
>
> Also, studies regarding the model's complexity concerning inference time on resource-limited devices have recently attracted attention.
>
> * Kuzmin, Andrey, et al. "Pruning vs quantization: which is better?." Advances in neural information processing systems 36 (2024).
>
> * Zheng, Hong-Sheng, et al. "StreamNet: memory-efficient streaming tiny deep learning inference on the microcontroller." Advances in Neural Information Processing Systems 36 (2024).
>
> These studies redesign models primarily based on FLOPs or inference time in seconds, so their analysis depends on the device's choice.
>
> In this regard, one of this study's key contributions and differences from previous works is improving model performance while maintaining the model's complexity based on theoretical time complexity.
>
> Because these aforementioned works are closely related to our work and were presented at the NeurIPS, we believe that our work is within the scope of the NeurIPS conference.

---

> > ### Comment · Reviewer_VmYr · 2024-08-14
> >
> > I agree with the authors that the work presented is of relevance to the NeurIPS community. This is well-justified by the relevant work that you've have cited.
> >
> > I would like to thank the authors for your timely and clear responses to my comments.
> >
> > My rating stands as 6 because the proposed work is specific to bio-signals instead of time series modelling in general.
> >
> > Good luck :D

---

### Official Review · Reviewer_1Z8p · 2024-07-12

**Soundness:** 2
**Presentation:** 2
**Contribution:** 2
**Rating:** 5
**Confidence:** 4

**Summary:**

This paper proposes an all-layer aggregation network, CALANet, to improve model accuracy while maintaining the efficiency of lightweight models. Specifically, the authors have exploited the features from all layers for classification, as a kind of aggregation.

**Strengths:**

1. The motivation of this work is clear. It sounds reasonable that the features from all layers can provide more information for classification, compared to merely using the features from the last layer.
2. Extensive experiments have been conducted, including comparison results and ablation study.
3. The organization of paper is clear and the writing is easy to understand.

**Weaknesses:**

1. My major concern is about the aggregation. When I first read the title of this paper, I assumed that the authors have trained a neural network and then compress its multiple layers via aggregation. However, it seems that the authors just add several components (LCTMS, SLAP) to a complete CNN. It looks like an advanced version of residual networks with channel weights.
2. The authors claimed that they wanted to improve model performance while keeping model efficiency. However, in Table, it seems that, in some cases (KU-HAR, PAMAP2), the proposed method did not have obvious F1 improvements but resulted in highly increased FLOPs.

**Questions:**

1. How to prove that the performance improvement is because of the aggregation, but not because of the more complicated network architecture? To be more specific, what if we do not use the proposed LCTMs and SLAP but directly increase the layers of CNNs? It would be convincing if the authors provide results with varying numbers of layers with/without LCTMs and SLAP.
2. The authors have used FLOPs to measure model efficiency, which may be inadequate, because the best hyperparameters for different models are different. If the authors directly use the same hyperparameters for all models, we may see a fair comparison in terms of efficiency, but it is not fair to compare their accuracy/F1. Can the authors provide the results in terms of training/testing time for intuitive efficiency comparison (in Table 1)?

**Limitations:**

1. This is not a model compression work but just proposed some aggregation modules. The authors should discuss the difference between their work and model compression works.
2. This work mainly focuses on CNNs. The authors can further discuss other network architectures.

---

> ### Author Rebuttal · Authors · 2024-08-07
>
> We are grateful for the constructive comments of the reviewer. Below, we provide specific answers and explanations regarding those comments.
>
> **(W1) My major concern is about the aggregation. When I first read the title of this paper, I assumed that the authors have trained a neural network and then compress its multiple layers via aggregation. However, it seems that the authors just add several components (LCTMS, SLAP) to a complete CNN. It looks like an advanced version of residual networks with channel weights.**
>
> **(L1) This is not a model compression work but just proposed some aggregation modules. The authors should discuss the difference between their work and model compression works.**
>
> Thank you for your constructive comments. There is an important difference between our study and model compression works. The objective of model compression is to reduce model size and computations while minimizing loss of accuracy. Conversely, our goal is to improve accuracy while maintaining computations. Thus, we did not focus on compressing the model. Instead, we fixed its complexity because the computational capability of wearable devices is fixed and hard to change.
>
> **(W2) The authors claimed that they wanted to improve model performance while keeping model efficiency. However, in Table, it seems that, in some cases (KU-HAR, PAMAP2), the proposed method did not have obvious F1 improvements but resulted in highly increased FLOPs.**
>
> **(Q1) How to prove that the performance improvement is because of the aggregation, but not because of the more complicated network architecture? To be more specific, what if we do not use the proposed LCTMs and SLAP but directly increase the layers of CNNs? It would be convincing if the authors provide results with varying numbers of layers with/without LCTMs and SLAP.**
>
> Thank you for your constructive comments. If what we have figured out is correct, we apologize for the confusion caused by removing a version without layer aggregation from our CALANet in Table 2 of the submitted paper. To avoid these confusions, we will add Figure 1 and Table 2 in the attached .pdf file to the paper.
> Please check the attached .pdf file in “Global(Author) Rebuttal”. Figure 1 shows the tradeoff between the FLOPs and F1-score with varying numbers of layers in CALANet with/without LCTMs and SLAP.
> The tradeoff curves closer to the top-left are more efficient, with a higher F1-score per FLOPs. As shown in Figure 1, CALANet with LCTMs and SLAP the higher F1-score in similar computational cost than one without LCTMs and SLAP.
>
> **(Q2) The authors have used FLOPs to measure model efficiency, which may be inadequate, because the best hyperparameters for different models are different. If the authors directly use the same hyperparameters for all models, we may see a fair comparison in terms of efficiency, but it is not fair to compare their accuracy/F1. Can the authors provide the results in terms of training/testing time for intuitive efficiency comparison (in Table 1)?**
>
> Thank you for your constructive comments. We are measuring the train and test times for all baselines and datasets in Table 1. In our measurements, we found that the testing time of the network architecture depends on not only FLOPs but also various environments, including devices, memory available space, and background apps. Interestingly, in most cases, it was confirmed that applying LCTMs and SLAP to a given specific architecture hardly increased the time.
> Also, the training time tended to be almost proportional to FLOPs because we used the same epochs for all models, and larger models used more memory accesses.
>
> **(L2) This work mainly focuses on CNNs. The authors can further discuss other network architectures.**
>
> Thank you for your constructive comments. In sensor-based human activity recognition (HAR) literature, popular architectures include CNNs, RNNs, and Transformer.
> Compared with CNNs, RNNs in terms of real-time HAR have some weaknesses as follows:
>
> 1) poor parallelization due to dependency of computations,
> 2) the lack of hardware accelerators for edge device deployment.
>
> Meanwhile, Transformer needs to calculate the relationship between timestamp of which each is meaningless.
> Furthermore, the hybridization of CNNs and Transformer can be promising approach. We want to emphasize strengths of CALANet because the self-attentions commonly are calculated for the output of convolutional layers due to their local temporal modeling capability. Therefore, our LCTMs and SLAP can improve the hybridization of CNNs and Transformer.

---

> > ### Comment · Reviewer_1Z8p · 2024-08-12
> >
> > Thanks for your detailed response. Good luck.

---

### Official Review · Reviewer_QGqB · 2024-07-12

**Soundness:** 3
**Presentation:** 3
**Contribution:** 2
**Rating:** 6
**Confidence:** 4

**Summary:**

The problem of Human Activity Recognition (HAR) is considered in this paper where the border between different activities can differ depending on the type of activity. For example, one activity can be “just sitting”, and another can be “sitting while speaking”. To this end, the authors argue that we need to leverage the features extracted in all the layers of a neural net. Thus, the authors propose a modification to ConvNet models where (called “all-layer aggregation”) is added to the model which takes its input from all the previous conv layers. To do so, the authors design learnable channel-wise transformation matrices that can be added to the model and provide fast aggregation. Evaluation results on seven HAR dataset shows that the proposed modification can improve the classification accuracy of HAR, compared to other alternatives.

**Strengths:**

This paper spots a very interesting problem in HAR and offers an effective solution with a nice architectural modification. The paper is written well and easily understandable.

**Weaknesses:**

The weakness of this paper is its relevance to the NeurIPS. This work gets more attention and appreciated by people in the Mobile or UbiComp community as the novelty and contribution is not much in the ML part but in systemic modification of the architecture.

**Questions:**

1. It seems that such a modification is only applicable to ConvNets. It is not clear how similar things can be done to other architectures.

2. I could not understand whether this CALANet is only useful to HAR or if it can also help with other data types like audio data or bio signals.

3. It would be interesting and useful if the authors could show how this modification can be applied to some benchmark architectures that are built for mobile or wearable devices such as MobileNet or EfficientNet, and similar models.

**Limitations:**

The main limitation is that the solution is very specific to HAR datasets and it is only applied to basic ConvNets and not other benchmark models.

---

> ### Author Rebuttal · Authors · 2024-08-07
>
> We are grateful for the constructive comments. Below, we provide specific answers and explanations regarding those comments.
>
> **(W1) The weakness of this paper is its relevance to the NeurIPS. This work gets more attention and appreciated by people in the Mobile or UbiComp community as the novelty and contribution is not much in the ML part but in systemic modification of the architecture.**
>
> Thank you for your constructive comments. We respect good opinions, but we believe that the discovery of new structures in architecture can also serve as a foundation for important research in the field of ML.
>
> **(Q1) It seems that such a modification is only applicable to ConvNets. It is not clear how similar things can be done to other architectures.**
>
> **(Q3) It would be interesting and useful if the authors could show how this modification can be applied to some benchmark architectures that are built for mobile or wearable devices such as MobileNet or EfficientNet, and similar models.**
>
> Thank you for your constructive comments. Some constraints must be satisfied to apply our method to other architectures effectively: 1) the layers of a network architecture should be calculated sequentially and independently; 2) the forward pass should include the resolution reduction operation like the pooling layer. 3) the output of each layer should be able to be expressed as a (temporal length * channel size) matrix.
> To the best of our knowledge, most wearable sensor-based human activity recognition models can satisfy the above constraints. For example, in Inception-like models, LCTMs and SLAP can be effectively applied for each module rather than each operator.
> Please check the attached .pdf file in “Global(Author) Rebuttal”. In Table 1, we applied our LCTMs and SLAP to SqueezeNet. Specifically, the output of a squeeze convolution layer in each fire module is fed into LCTMs and connected to the last layer.
> As a result, our modification significantly improved F1-score of SqueezeNet on 71% of the all datasets while maintaining its FLOPs.
>
> **(Q2) I could not understand whether this CALANet is only useful to HAR or if it can also help with other data types like audio data or bio signals.**
>
> Thank you for your constructive comments. CALANet is useful when information lost in the intermediate layer affects classification accuracy. In this regard, our CALANet can be useful to some applications using bio signals. On the other hand, most audio data is collected in high sampling frequency. Therefore, these applications require the capability modeling long-term temporal dependency. In this regard, CALANet may not be appropriate.

---

> > ### Comment · Reviewer_QGqB · 2024-08-09
> >
> > Thank you for your response and enthusiasm for improving this work. Considering additional datasets, such as biosignals similar to motion sensors, can improve the demonstration of this work's applicability to other ML applications. Also, a proper discussion on how the method presented in this work can be applied to different architectures is necessary. The new results should also be appropriately integrated into the paper.

---

> > > ### Author Response · Authors · 2024-08-14
> > >
> > > Thank you for your recommendation and encouragement.
> > >
> > > We applied CALANet to the ECG heartbeat classification problem using the MIT-BIH arrhythmia dataset [1], which includes 24-hour ambulatory ECG recordings collected from inpatients and outpatients at Boston's Beth Israel Hospital. The dataset has 21,892 heartbeats, each with a signal length of 187. In the following Table, CALANet exhibited comparable performance with other networks designed to process ECG signals. This result shows that CALANet has promising applicability to other ML applications.
> > >
> > > ||CALANet|Pham et al. [2] |Ganguly et al. [3]|
> > > |:---|:---:|:---:|:---:|
> > > |Average Accuracy|98.2|98.5|97.3    |
> > >
> > > In addition, we elaborate on how CALANet can be applied to other architectures.
> > >   1. Define the smallest unit of a set of adjacent operators repeated across the network architecture as the "layer," such as the fire module of SqueezeNet and the residual module of ResNet.
> > >   2. Check if the output of each layer can be expressed as a (temporal length $\times$ channel size) matrix.
> > >   3. Multiply the output matrix and LCTM for each layer, and concatenate the output vectors across layers.
> > >   4. Feed the concatenated features into the classifier to predict activity.
> > >
> > > > [1] Goldberger, Ary L., et al. "PhysioBank, PhysioToolkit, and PhysioNet: components of a new research resource for complex physiologic signals." circulation 101.23 (2000): e215-e220.
> > >
> > > > [2] Pham, Bach-Tung, et al. "Electrocardiogram heartbeat classification for arrhythmias and myocardial infarction." Sensors 23.6 (2023): 2993.
> > >
> > > > [3] Ganguly, Biswarup, et al. "Automated detection and classification of arrhythmia from ECG signals using feature-induced long short-term memory network." IEEE Sensors Letters 4.8 (2020): 1-4.

---

### Author Rebuttal · Authors · 2024-08-07

# General Response

We thank the reviewers for their detailed feedback and valuable comments. We are glad the reviewers find that
- Our paper deals with a novel and interesting question and has a clear motivation.
  - "This paper spots a very interesting problem in HAR and offers an effective solution with a nice architectural modification" – QGqB
  - "The motivation of this work is clear" - 1Z8p
  - "CALANet addresses an important question for efficient mobile computing in the HAR space" – VmYr
  - "It was cool to see the authors start the paper with an empirical observation on how the features at different layers might differ to motivate the work" – VmYr
  - "The topic of sensor-based HAR is interesting and important, aligning with the scope of the ML community" - 2n6z
  - "It is good to see future studies are also discussed" - 2n6z
- Our claim and approach are theoretically reasonable.
  - "It sounds reasonable that the features from all layers can provide more information for classification, compared to merely using the features from the last layer" - 1Z8p
  - "This paper presented both theoretical guarantees and empirical evidence to validate the computational efficiency of the proposed model" – VmYr
  - "I appreciate authors for providing theoretical proofs, for the complexity and others" - 2n6z
- Our paper is well-organized and easily understandable.
  - "The paper is written well and easily understandable" – QGqB
  - "The organization of paper is clear and the writing is easy to understand" -1Z8p
  - "The manuscript reads well" – VmYr

We agree that some aspects of the paper can be improved, and many suggestions will be incorporated in the paper. We respond to individual comments below but briefly provide some common responses here. If any questions are unanswered or our responses need clarification, we would appreciate the chance to engage further with our reviewers.

One of the key concerns that multiple reviewers raised during the review process was that the experiments and performance analysis needed to be improved. Attached is a file containing the additional experiments suggested by the reviewers.
1. Reviewer QGqB asked about whether our architectural modification can be applied to some benchmark models that are built for mobile devices. Table 1 in .pdf file shows the comparison results of SqueezeNets with/without our cheap all-layer aggregation (LCTMs and SLAP). We explain in detail in our response below.
2. Reviewer 1Z8p asked if the authors provide results with varying numbers of layers with/without LCTMs and SLAP. Figure 1 in .pdf file shows the tradeoff between the FLOPs and F1-score with varying number of layers; here, tradeoff curves closer to the top-left are more efficient, with a higher F1-score per FLOPs. We explain in detail in our response below.
3. Reviewer VmYr suggested the ablation studies using the same depth of networks and probably similar FLOPs across different baselines. Table 1 in .pdf file compared two networks with similar FLOPs and with/without our cheap all-layer aggregation. In addition, Figure 1 shows an efficiency of our layer aggregation. We explain in detail in our response below.
4. Reviewer 2n6z suggests that we compare CALANet and more advanced models. We conducted an additional experiment with two larger models in Table 2. We explain in detail in our response below.

Once again, we are grateful for the time and effort put into reviewing this submission, and we firmly believe that these comments will strengthen the clarity of our manuscript.

---

### Decision · Program_Chairs · 2024-09-25

**Decision:**

Accept (poster)

**Comment:**

The paper considers human activity recognition (HAR) from wearable IMU sensor measurements which has been very active research area
more than a decade. Paper is focusing on improving deep learning -based HAR by proposing convolutional neural network utilizing novel
architecture to improve the recognition accuracy and decreasing the computational complexity and size of the network for real-time small resource (on the edge/mobile devices) inference. Proposed approach is based on all-layer feature aggregation with learnable channel weight matrices and fast fusion. Proposed methodologies are empirical evaluated against several related methods on publicly available HAR benchmark datasets, showing improvement of recognition accuracy meanwhile reducing the number of computational operations.

Although in borderline, this paper has received positive feedback, and based on scores most of the reviewers are towards accepting the paper. The strengths of the paper are related to motivated and interesting topic with clear research goal (Reviewers QGq8, 1Z8p, and  2n6z), good empirical evaluation (Reviewers 1Z8p and 2n6z) with proper theoretical analysis of background (Reviewers VmYr and 2n6z), and promising results against other models on benchmark datasets. Weaknesses/limitations are related to focusing only on narrow area of HAR (Reviewers QGqB and VmYr), some limitations on evaluation and ablation study (Reviewers VmYr and 2n6z), and not well-justified usage of CNN architecture (e.g., compared to RNN/transformers, pointed by Reviewer 2n6z).

Authors have provided response to critics in rebuttal by showing some additional experimental result, including: 1) additional ECG-classification task with comparable accuracy against baselines (in discussion comments with Reviewer QGqB), 2) additional ablation study showing usefulness of LCTM and SLAP (in global rebuttal file, Fig 1/ Table 1), and 3) additional experiments against transformer based architectures (in discussion comment with Reviewer 2n6z). Although somewhat narrow application area where the method is proposed to, it has few important merits (described above) and brings interesting knowledge to NeurIPS community (in addition to Ubicomp/wearable computing domain). To keep conference diverse, I am recommending the paper to be accepted as a poster. Authors are encouraged to revise the final paper, including all the improvements and suggestion raised during the rebuttal discussion.